# Testing the Environmental Kuznets Curve Hypotheses in Chinese Provinces: A Nexus between Regional Government Expenditures and Environmental Quality

**DOI:** 10.3390/ijerph18189667

**Published:** 2021-09-14

**Authors:** Ayoub Zeraibi, Daniel Balsalobre-Lorente, Khurram Shehzad

**Affiliations:** 1School of Economics and Finance, Xi’an Jiaotong University, Xi’an 710049, China; 2Department of Political Economy and Public Finance, Economic and Business Statistics and Economic Policy, University of Castilla-La Mancha, 13001 Ciudad Real, Spain; Daniel.Balsalobre@uclm.es; 3School of Economics and Management, Southeast University, Nanjing 211189, China; khurramscholar64@hotmail.com or

**Keywords:** China, government spending, environment’s sustainability, N-shaped, orthogonal–GMM

## Abstract

With rapid economic growth, the Chinese government expenditures at various levels have increased adequately. At the same time, the environmental quality in China has deteriorated significantly. In this study, provincial-level data for 31 Chinese provinces during 2007–2017 are used to investigate the impacts of government expenditure on the emissions of three specific measures of environmental degradation. The main objective of this study is to examine the influence of government expenditures, economic growth per capita, environment protection expenditure, and added second-sector value on environmental quality by measuring sulfur dioxide (SO_2_), chemical oxygen demand (COD), and ammonia nitrogen emissions (AN). Moreover, the study applied the generalized method of moments (GMM) and the fully modified least square (FMOLS) to estimate the co-integration relationship among the underlying factors. The results demonstrate a significant direct effect of government expenditure on improving environmental quality overall in the Chinese provinces, which increases with the level of economic growth. However, the results also confirmed the inverted N-shaped relationship between the pollution factor and economic growth per capita. Our key findings lead toward the manifestation and emphasis of the importance of appropriate policies for restoring government expenditure and, at the same time, strengthening the relationship between the industrial sector and environmental policy standards. Significantly, governments in developing countries should allocate larger budgets for environmental projects in their fiscal reforms for the sake of moving to greener and more inclusive economies with low-carbon activities.

## 1. Introduction

Due to economic growth, environmental pollution and protection have grown increasingly severe in many developing countries, particularly China. As the world’s second-largest economy, the Chinese government has invested much in environmental regulation and protection, 37 trillion USD. Moreover, China has invested 41 trillion USD in their 18 provinces for the treatment of water pollution. In addition, China is introducing energy-efficient and environmentally friendly products to overcome environmental pollution [1]. However, China ranked 14th in environmental pollution throughout the world [2]. Hence, air pollution has become one of the most pressing issues affecting people’s health and China’s economic development in the new millennium [3,4]. The problem of environmental pollution in developing countries is increasingly diminished as economic growth takes hold in countries such as China. This process is known as the composition effect [5,6]. In recent years, various studies have examined the effect that government expenditure and financial development have on environmental quality in regions such as the Middle East [7]. The concept of the scale effect comes about through how government spending, which is considered an income effect, increases when environmental pollution occurs due to economic growth.

Having one of the world’s largest economies, the Chinese government has made efforts to reduce environmental pollution caused by rapid economic growth by imposing regulations limiting the amount of pollution caused by commercial industries. Environmental quality can be improved by utilizing human capital more than physical capital since it is more environmentally friendly, increasing government spending. Some studies mainly focus on economic and social determinants such as GDP per capita and urbanization [8], while other studies examine how government policies influence environmental pollution [9].

Some authors [9,10] have investigated how CO_2_ emissions are influenced by environmental innovation and the effects of decentralization on environmental quality in China. Such studies, amongst others, give evidence of how government expenditure is a substantial factor [7]. In Northern Africa and the Middle East, environmental quality is affected by public expenditure, as shown in [9].

Using data from 77 countries, the study by [7] determined that CO_2_ and SO_2_ emissions have a negative effect on government expenditure. Similar to [11], they argued that as the economy grows, environmental pollution also increases, increasing government spending, which is namely, the scale effect. In contrast, the composition effect utilizes human capital over physical capital, which can be harmful to the environment. Furthermore, this can lead to what is known as the technique effect, whereby government expenditure increases due to the increase in labor productivity.

Government expenditure can also increase through the income effect; as more people obtain a higher income, their care for better environmental quality is also enhanced. The study by [12,13] shows the impact that government expenditure has on environmental quality, and it can differ according to some influential determinants or pollutants.

In this study, three pollutants are used in the investigation to determine the effect that government spending has on environmental quality in China. This study uses three main determinants for the investigation: SO_2_, COD release, and AN emissions. These pollutants are used because they have different characteristics and can provide a detailed environmental quality analysis. They have been under the government’s control for longer due to their harm to ecosystems and the environment [13]. SO_2_ emissions have contributed to the pollution problem in China in recent years. This is also true for COD release that indicates water pollution due to coal combustion [14]. China planned and managed to significantly reduce COD release by 8% within a period of six months through the 12th Five-Year Plan.

Figure 1a–d, indicates the three pollutants in the 31 provinces of China (SO_2_, COD release, and AN emissions levels) and regional economic growth per capita. We observed in Figure 1 that the provinces with a high level of economic growth are also offset by a high level of pollution, except for the provinces of Beijing and Shanghai, whose local governments are working to raise the volume of spending on the environment and improve its quality [15].

We also note that dioxide emissions have increased in the northwestern and northeastern regions, which are areas rich in mining and metallurgical industry [16]. However, these regions have a low growth rate because the region is for the extraction of raw materials and is far from the coast and major commercial places in China. Hence, the extraction of raw materials contributes to rising levels of pollution.

In Figure 2, the graph shows the provinces of China that have government spending to protect the local environment. The figure shows that there are provinces that are interested in preserving the environment by increasing the volume of government expenditures on the environment, such as Beijing, Jiangsu, and Guangdong, similar to the Hubei region, which is famous for its heavy industries with high pollution. This study’s essential contribution is two-fold. First, the current study is one of the few that analyze the relationship between sources of pollution and economic growth in the context of the impact of government expenditure and environmental quality under an N-shaped EKC scheme. Second, this study measures the effects of three sources of emissions—COD release, SO_2_, AN emissions—on economic growth in various provinces of China [17]. This study also explores the total local government expenditure contribution, taking into account the local government spending to protect the environment in Chinese provinces. Previous studies have used carbon emissions to measure environmental pollution. However, only carbon emissions alone cannot characterize environmental pollution in its entirety [18]. Hence, this investigation employed three proxies, SO_2_, COD, and AN emissions, which measure maximum environmental pollution. Particularly, COD release is a significant measure of water pollution, which is used by various studies. Consequently, the current study allows us to measure the extent and seriousness of the Chinese government’s contribution towards preserving the environment in its general form on the Chinese state as a whole and also in part, represented by the Chinese provinces [19]. This is in contrast to previous studies, which relied on short- and long-term analysis in their approach [20]. The remainder of this paper is developed as follows: The relevant literature review is conducted in Section 2. The data and methodology are introduced in Section 3. The estimation results and analyses are described in Section 4. The conclusion appears in the last section, which also provides several policy recommendations.

## 2. Literature Review

To date, there have been several studies on the factors that influence environmental pollution. Various studies, such as the Kuznets environmental curve and the hypothesis of a pollution refuge, are based on traditional theories explaining the relationship between environmental quality and economic growth [21,22]. Several previous studies have investigated the relationship between government expenditure and environmental quality in China. However, the results diverged significantly. The use, and then the excessive use and mitigation of pollution, and climate change, make up a broad field of empirical research in the literature pertaining to economic growth and the environment [23]. Moreover, the authors explored the linkage between the government expenditure and pollution factors in China by using the GMM approach for the period 2002–2014; the study result indicates the existence of the EKC hypothesis with an inverted U-shape and N-shape. A similar study by [24] analyzed the relationship between SO_2_ emissions and government expenditures in the case of China for the period 2008–2013, which applied spatial competition; the result confirmed a positive spatial correlation of SO_2_ emissions among provinces in China. It also confirmed the existence of the N-shaped EKC hypothesis. The examination also indicated a positive association between the second-sector value and SO_2_ emissions.

Similarly, [25] clarified the relationship between environmental quality and economic growth in the context of the EKC hypothesis, and used SO_2_, emissions together with COD release. The investigation result indicated a relationship between economic growth per capita and SO_2_ emissions and COD release in the period 1998–2016, which also confirmed the pollution halo hypothesis. In the same context, [26] examined the association relationship between noise pollution and economic growth in 111 Chinese cities by using the GMM approach to test the existence of the N-shaped EKC hypothesis; the result confirmed the invert N-shape between economic growth per capita and noise pollution.

In contrast, [27] investigating the linkage between the fiscal expenditure and SO_2_ emissions as an environmental quality proxy in China at the provincial level, in the period 1995 to 2015, applied the GMM approach to determine the role of provincial and regional economic growth on financial autonomy in China [28]. The results indicate an inverted-U shape between economic growth per capita and fiscal expenditure [29]. The study also shows a positive relationship between economic growth and fiscal expenditure, together with an inverted U-shaped and N-shaped relationship between the SO_2_ emissions and economic growth in China [30]. Moreover, the study also indicates that the second-sector value adds positively to affect environmental degradation. In contrast, explored the relationship between the local government expenditure and the SO_2_ emission in 31 of China’s provinces from 2007 to 2016. The econometric approach reveals a negative relationship between SO_2_ emission and local government expenditure. In addition, the study indicates the existence of an inverted U-shaped EKC hypothesis in a few provinces and not in others [31]. The relationship between the fiscal government and environmental quality in the top ten Asian carbon emissions for the period 1981–2018 was analyzed in [31]; the result shows that positive shock in government expenditure will worsen environmental quality in Malaysia, UAE, Thailand, Indonesia, Turkey, Iran, India, and China, but improve it in Japan. In the same contexte, [32] applied the GMM method to examine the relationship between SO_2_ emissions, COD release, and AN emission pollution factors and local and provincial economic growth in 31 provinces in China for the period 2004 to 2016. The study’s result indicates the positive relationship between economic growth and the three pollutions factors. This corresponds to [33], who clarified the linkage between government expenditure as fiscal policy proxy and environmental quality in China for the period 1980–2016. The empirical result indicates the positive relationship between government expenditure and environmental degradation with the existence of EKC hypothesis, both in the long and short term. The analysis by [34] analyzed the linkage between SO2 emission and regional economic growth in 29 provinces for the period 1998–2015. The empirical results indicated a positive relationship between the regional economic growth and the pollution factor the SO2 emission, validating the inverted-U and the N-shaped EKC hypothesis. Moreover, in the context of Pakistan [35], when investigating the nonlinear relationship between the role of fiscal policy on environmental quality, the empirical result shows the negative effect of fiscal policy instruments on the environmental quality in the long term, with the validation of the inverted U-shaped relationship for the period 1985 to 2016. However, the analysis by [36], when analyzing the effect of both fiscal expenditure and environmental expenditure on the environmental quality in the emerging economies for the period 1995 to 2016, indicated the positive effect of the environmental expenditure and environmental degradation and negative effect on the fiscal expenditure. Additionally, the empirical analysis by [37] clarified the nexus between public expenditure and environmental quality in 94 countries, which measured the environmental quality by the SO_2_ and AN emissions for the period 1970 to 2008. The study demonstrated the significant positive and direct effect of public expenditure on the environmental quality factor. The total effect was divided into direct and indirect effects, and both short- and long-term impacts were discussed. In this research, the environmental Kuznets curve (EKC) hypothesis model was used to examine the impact of government expenditure on pollutant emissions. 

Thereafter, many researchers investigated the existence of an EKC hypothetical relationship between GDP per capita and certain pollutants [16,17]. However, the estimation results are still controversial [38]. Found evidence for the existence of an inverted N-shaped EKC hypothesis for COD release for all of China and Shanghai city. In addition, the existence of an inverted U-shaped EKC in the G20 group members was reported.

Additionally, [12] analyzed Chinese survey data and found that the relationship between CO_2_ emissions per capita and GDP per capita is N-shaped, suggesting there may not be a turning point for CO_2_ emissions in China. As many previous studies have used typical pollutants as the proxy of environmental quality, in this study, three representative pollutants SO_2_ emissions, COD release, AN emissions—are chosen for empirical estimations. These contrasting pollutants have various characteristics and therefore reflect the different aspects of China’s environmental pollution. Consequently, examining these pollutants helps us understand the full story of environmental impacts related to government spending in China. 

Despite several previous studies, the link between government expenditure and environmental quality in China has been rarely explored, and mainly only when using provincial data. As a result, this research analyzes panel data from 31 Chinese provinces to examine the link between government expenditure and SO_2_ emission, COD release, and AN emission pollutants; furthermore, the total effect of government expenditure is estimated. 

## 3. Methodology and Data Description

### 3.1. Methodology

This study applies the generalized method of moments (GMM) methodology to determine provincial government expenditure’s direct and indirect impacts. The study’s econometric approach is the generalized method of moments (GMM) technique. The GMM approach is used to determine the direct and indirect impacts of provincial government. Since the goal of the research is to distinguish the direct and indirect effects of the proportion of government expenditure over GDP, the estimation method utilized in this research is a two-stage regression model. First, under the EKC framework, this study examines the impact of government expenditure on GDP per capita, applying both the GMM and data fixed effects methods. The GMM approach was introduced by [39] then developed by [40]. Previous studies have used this method to analyze the connection between GDP and pollutant emissions [41]. The results obtained from this research using the GMM method are used as the benchmark due to its efficiency in analyzing the data in different Chinese cities. Moreover, the first-difference GMM method can solve the endogeneity problem due to bilateral causality between GDP per capita and government spending [42].The impacts of government expenditure on pollutant emissions are estimated using the environmental Kuznets curve [43]. The EKC hypothesis explains an inverted N-shaped relationship between economic growth and environmental degradation; environmental pressure increases in the early stages of economic growth due to the increased release of pollutants and the extensive and intensive exploitation of natural resources associated with the greater use of production resources, and the adoption of certain production technologies for the growing economic activities, up to a certain level, as income rises [44,45]. After that, it decreases, probably because of the growing public awareness and concern about environmental degradation and the research and development activities being oriented more toward the concept of the green economy when GDPP at a high level. The third stage, which can return to the parallel relationship between economic growth and pollutants, implements the cube of economic growth.

Both the fully modified ordinary least square (FMOLS) and the generalized method of moments (GMM) approach are used in the empirical estimations. introduced the generalized method of moments (GMM), which was studied further by others [46]. Because it can eliminate any endogeneity bias, many previous studies have used the GMM method to estimate the relationship between GDPP and pollutant emissions. The results of the first-difference GMM technique are used as the benchmark in this study because individual differences may be eliminated [47]. It is reasonable to infer that there are significant differences between Chinese provinces, which the explanatory variables can highlight.

The following equation is used in the first step of estimating the direct effect of go share on GDP per capita:(1)$$\left(LnGDPP_{it}\right)=\mu_{i}+\delta_{t}+\alpha_{0}\left(LnGDPP_{it}\right)+\beta_{1}\left(LnLGE_{it}\right)+\beta_{3}\left(LnENVE_{it}\right)+\beta_{4}\left(LnLSSVA_{it}\right)+\varepsilon_{it}$$

In Equation (1), ln is the logarithmic term for the province I’s GDP per capita in year t. (1.) *GDP_it_*. In addition, a lag factor was introduced to the factor to account for the link between annual GDP per capita and pollutant emissions factors: COD release, SO_2_, and AN emissions [48]. The logarithmic terms of province government expenditure on the environmental and second-sector value added are given by the control variables ENVE and SSVA, respectively. The second equation examines the direct effect of go share on pollutant emissions, the same as Equation (1) estimates the government proportion on GDPP:(2)$$\begin{array}{c} \left(Ln\gamma_{it}\right)=\theta_{i}+\sigma_{t}+\left(Ln\gamma_{it}\right)+\beta_{1}\left(LnGDPP_{it}\right)+\beta_{2}{\left(LnGDPP_{it}\right)}^{2}+\beta_{3}{\left(LnGDPP_{it}\right)}^{3}+\beta_{4}\left(LnLGE_{it}\right)\\ +\beta_{5}\left(LnENVE\right)+\beta_{6}\left(LnSSVA_{it}\right)+\varepsilon_{it} \end{array}$$

The $\left(\gamma \right)$ stands in different pollutant emissions per capita, such as COD release, SO_2_, and AN emissions; ENVE represents the local government environmental expenditure; SSVA represents the second-sector value added; moreover, the study variables are in logarithmic terms. According to the Equation (2), the direct and indirect effects can be calculated. Specifically, the direct effect is merely a constant number, such as the coefficients of β4 in Equation (2), while the indirect effect is the product of the marginal effect of go share on GDPP and GDPP on pollutant emissions. 

### 3.2. Data Description

This study used data from 31 Chinese provinces between 2007 and 2017. The selected panel data were used because of its larger sample size and capacity to efficiently handle heteroskedasticity across the cities efficiently. The data characteristics include the province’s number (N = 31), which is larger than the study period (T = 20). Metropolitan regions produce the most pollutants; therefore, municipal statistics might offer a detailed picture of the environmental status. The Chinese government’s interest in the environment and climate change has increased, which led it to allocate spending at the county level to protect the environment. The relationship between government expenditure and environmental quality are examined and investigated by introducing the second stage of the secondary industry value-added to the GDP per capita [49]. We used three variables chosen as the main variables; first, two wastewater pollutants: chemical oxygen demand emissions (COD) (10,000 tons) and ammonia nitrogen emissions (AN) (10,000 tons) in wastewater; second, the study tested the emission of gases and their impact on environmental degradation, sulfur dioxide emission (SO_2_) (10,000 tons) in waste gas was used as the variable.

We used the proportion of provincial government expenditure, provincial environmental expenditure, and the second-sector value added using the proportion of provincial government expenditure, provincial environmental expenditure, and second-sector value added as independent variables. General budget expenditure of local finance amounted to (100 million CNY). The most relevant variable is the provincial government expenditure (100 million CNY), computed by subtracting annual general provincial government expenditure when these variables were collected from the National Bureau of Statistics of China. Moreover, the study variables were mentioned below in Table 1.

Table 2 implicitly shows the descriptive statistics of the underlying variables. Table 2 confirms that COD release, SO_2_, and AN emissions in China have maximum values of 5.289, 10.416, 3.139, and minimum values of 0.405, −1.966, −1.388, with an average of 3.738, 7.362, and 1.388, respectively. This illustrates a large disparity in environmental pollutant emissions in the different provinces of China. Problematic and threatening to the environmental system in China, Table 1 also depicts the features of government outbursts when there is a large discrepancy between the minimum and maximum values and a very large standard deviation of 0.758796, which indicates that government expenditure varies greatly across Chinese regions. However, the average value of government expenditure is 7,881,049 CNY across Chinese provinces. The record of government expenditure (ENVE) on the environment sector and the added value of the second industrial sector (SSVA) shows an average value of 4.302 and 4.698, with a minimum value of 1.562 and 4.472, and a maximum value of 6.129 and 4.819, respectively. This characteristic is also reflected in the record of per capita GDP with an average value of 9.336, minimum value of 5.840, and maximum value of 11.425. However, there is a significant disparity in per capita GDP in different Chinese sectors. This explains the unevenness of economic growth in various Chinese regions.

## 4. Empirical Results

### 4.1. Estimation Results and Discussion

According to our hypothesis and the theoretical framework, an N-shaped relationship between income, when indicated by locale GDP per capita and capture of environmental degradation, should be expected in the estimations [50]. However, as seen in the Table 2, Table 3, Table 4 and Table 5, the results are inconclusive between classifications and the different methods used in the Fully Modified Least Square (FMOLS), difference– The generalized method of moments (GMM). There are two stages to the estimating technique [51].

The first—difference GMM approach—is the benchmark and the main method used. We utilized the (FMOLS) approach to evaluate the estimations’ robustness and confirm our hypothesis of the N-shaped EKC hypothesis for the total sample, the 31 China provinces [52]. In addition, the results of the three tests demonstrated the existence of an inverse relationship in the form of an N-shaped EKC hypothesis; we further discuss the result below [53].

Table 3 presents the outcomes of the relationship between the environmental qualities measured through three pollutant emissions factors, SO_2_, COD, and AN emissions, and provincial economic growth. The findings of FMOLS stated that the local government expenditure, SO_2_, COD, AN, and SSVE positively affect the GDP [54]. However, as per orthogonal–GMM and difference–GMM, SO_2_ and AN reveal a negative association with GDP. Moreover, environmental expenditures indicate a negative relationship with GDP; in addition, the result of the GMM–difference test also demonstrates its significance [55]. On the other hand, the results of FMOLS, orthogonal–GMM, and difference–GMM indicates a negative relationship of ENVE with GDP having a coefficient of −0.145, implying that one unit increase in ENVE will decrease 0.145 units of GDP. 

Table 4 enumerates the findings by taking the SO_2_ as the dependent variable. The results of FMOLS revealed that ENVE and SSVA have a positive link with SO_2_ while LGE reported a negative correlation with SO_2_. Moreover, difference–GMM and orthogonal–GMM denote the indirect impact of ENVE with SO_2_ emissions and the direct impact of LGE and SSVE on SO_2_. The study stated that an inverted N-shaped connection exists among the GDP, GDP^2^, GDP^3^, and SO_2_. The R-square and adjusted R-square value highlighted that 44.1% of the changes are due to the factors employed in this investigation. In comparison, 38.2% of changes occur due to significant factors among them. 

Based on analysis of the relationship between the pollutant emissions captured by SO_2_, COD and AN emissions, and regional economic growth, Table 3, Table 4, Table 5 and Table 6 clarify the relationship between the COD release, SO_2_, and AN emissions and the local government expenditure. In Table 5, we used the COD as a dependent variable when the FMOLS test indicates the significance of GDPP, GDPP^2^, GDPP^3^ at 1% level, and LGE and SSVA at 5%, and ENVE at 10%. The FMOLS test indicates the negative relationship between local government expenditure and chemical oxygen demand COD in China’s provinces, with a coefficient of 4.52%. The examination indicates a positive relationship of the environmental expenditure ENVE and SSVA with COD emissions, with a coefficient of 0.249 and 2.361, respectively. In contrast, the study confirms the existence of an inverted N-shaped hypothesis by the significance of GDPP, GDPP^2^, and GDPP^3^. The difference–GMM and orthogonal–GMM test followed the previous FMOLS result that shows stationary LGE, SSVA, GDPP, GDPP^2^, and GDPP^3^ at the first level. This also confirms the existence of the inverted N-shaped hypothesis when the difference–GMM shows a negative significance of GDPP, by 22.0%, and a positive significance of GDPP^2^, and again the GDPP^3^ indicated negatively on the COD release, with a coefficient of 0.1234%.

Table 5 exhibits the findings of the study when ammonia nitrogen AN is the dependent factor. The empirical findings confirmed the existence of an inverted N-shaped relationship between economic growth and AN. The difference–GMM and orthogonal–GMM showed that GDPP and GDPP^3^ have a negative coefficient, while GDPP^2^ presents a positive sign. FMOLS outcomes also verify these findings. Moreover, the study indicates the negative effect of local government expenditure on the AN emissions with a coefficient of –0.40459. The R-squared indicates that 66.4% variation in the AN occurred due to the variables used in this study. However, 65.8% changes in the AN are due to the significant variables among them. 

### 4.2. Discussing the Results of the Study

Based on Table 2, Table 3, Table 4, Table 5 and Table 6, the econometric results show consistent findings for any classification regarding the association between economic growth per capita and environmental degradation [56]. Although the FMOLS test showed an N-shaped EKC hypothesis for the capture of pollution nexus between the three pollutant emissions factors (COD release, SO_2_, and AN emissions) and provincial economic growth in the Chinese provinces, only a few quantiles confirm these results [53]. These inconclusive findings could depend on the heterogeneity between and within these revenue groups. A further breakdown of the provinces included and their specific characteristics, such as local development, the composition of industries, and local environmental laws, might be needed to fully understand why the N-shaped EKC is only apparent in some quantiles [57]. The current study was not presented in a way similar to the previous studies, by dividing the Chinese provinces into groups to avoid statistical problems and the small size of the sample. The first finding is an inverted N-shaped relationship between pollution variables and China’s province per capita income [58]. As per capita wealth rises, pollution variables decrease at first, then increase, and then decrease. As the economy progresses, the marginal advantage of regionally sound environmental quality has gradually increased [59]. By introducing rules and regulations to compensate for market failure, the government will respond to public demand for a clean environment and prevent pollution from increasing. As per capita income grows in the first stage, then pollution factors decrease as a result. New types of pollution will arise due to economic expansion, and existing market mechanisms and environmental regulations may not effectively mitigate these new types of pollution [26]. When the government has sufficient pollution data and uses cleaner manufacturing technologies, then it will set higher environmental standards and enforce stricter environmental rules [60]. The second turning point of the inverted N-shaped curve will occur simultaneously; the quality of the regional sound environment will improve even though both the FMOLS and the GMM estimations show an inverted N-shape [61].

## 5. Conclusions and Implications

The current study sheds light on the role that government spending generally played and local government spending specifically played in protecting the environment and improving environmental quality in 31 Chinese provinces between 2007 and 2017. It examines the nature of the relationship and the extent of the impact of local government spending on three variables that were chosen to measure the extent of pollution—COD release, SO_2_, and AN emissions—combining the contribution of both local economic growth and value added by the second sector, which measures the industrial strength of the Chinese provinces. Moreover, due to the nature of the data available, both the economic model FMOLS and GMM are applied. This study explores the nature of the relationship between the study variables, ensuring the EKC hypothesis. However, the study results agreed with the study’s objectives and hypotheses, as the analysis results show that the EKC hypothesis is fulfilled. In each of the first, second, and third models, an inverted relationship is identified between environmental degradation, COD release, SO_2_, AN, and economic growth in 31 provinces of China.

However, the study indicates that government expenditure is one of the determinants of pollution in China, including the local government expenditure on the protection of the environment. The econometric results reveal a negative connection between the capture pollution and GDPP, a positive relationship with GDPP^2^, and a negative relationship with GDPP^3^. According to the GMM test, we find a negative relationship between the government expenditure and capture of pollution. However, GDPP on environmental quality affected the local government expenditure indirectly.

Additionally, the GDPP, COD release, SO_2_, AN emissions decrease substantially, but at various rates and times. This paper has elucidated future studies in this field to explore the link between government expenditure and environmental quality in China using rigorous econometric approaches and splitting the overall effect into direct and indirect impacts. Several straightforward policy implications that may be inferred from these findings follow. For example, two components of government expenditure on environmental quality may be investigated further. Government expenditures on education, research and development (R&D), and infrastructure investment, for example, may all have an impact on the environment. Because SO_2_ emissions negatively influence people’s health, the government improved air quality previously. In contrast, COD release is not taken as seriously. As a result, recent years have seen an increase in emissions. Increased government spending can significantly minimize COD release and SO_2_ and AN emissions since a larger ratio of government expenditure decreases pollution emissions.

Based on the conclusions and preliminary results, we can give the following recommendation first: increase attention on the local environmental system of the Chinese provinces and work toward designing a system that balances local economic growth and environmental preservation. It is also necessary to work on managing local regulations and laws to cover various local environmental dimensions and sustainability, such as gaseous and industrial emissions and water pollution resulting from sewage. Second: work on building a local government spending system on foundations based on governance and monitoring the independence of government spending for local districts, which would maximize the volume of environmental conservation and reduce emissions. Third: improve the industrial structure and accelerate the transformation of the local industrial structure into an environment-friendly system. Although secondary industry was a significant growth engine for China’s economy, it caused growth of environmental pollution, which requires the intervention of the Chinese government to develop the industrial sector and achieve sustainable environmental development goals. Fourth: decision-makers should build strategies to preserve the environmental system on the most basic assumptions that keep pace with the size of the Chinese economy and the environmental challenges it faces, such as the Keynesian hypotheses. These depend on the cube of economic growth, and predict the existence of a hypothesis for the existence of a third stage, which allows for a return of the parallel between economic growth and environmental pollution, resulting in an inverted N-shaped relationship. This is in contrast to the classical EKC hypothesis, which stops at squaring the volume of economic growth, and explains the inverse relationship between economic growth and the importance of pollution, giving an inverted U-shaped relationship. This study encountered some limitations and provides guidelines for future research: (1) this study is limited to 31 Chinese provinces that have experienced further economic growth. (2) This study uses data from 2007 to 2017; the period for future studies can be extended by later studies. (3) In addition, future studies could also include an interactive term for energetic innovation and public investments in the environment. (4) Finally, the role of renewable energies and the governance criterion can also be covered in future studies.

## Figures and Tables

**Figure 1 ijerph-18-09667-f001:**
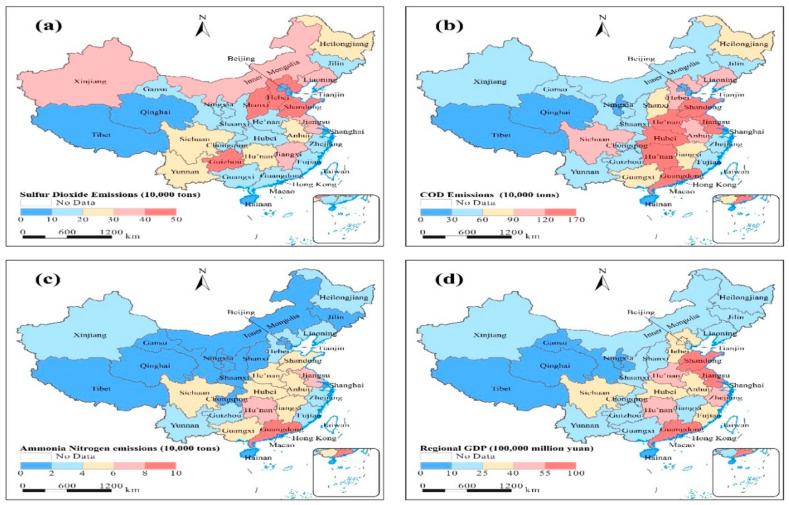
(**a**) Pollutants sulfur dioxide (SO_2_), (**b**) chemical oxygen demand (COD), and (**c**) ammonia nitrogen emissions (AN), and (**d**) provincial economic growth per capital (GDPP).

**Figure 2 ijerph-18-09667-f002:**
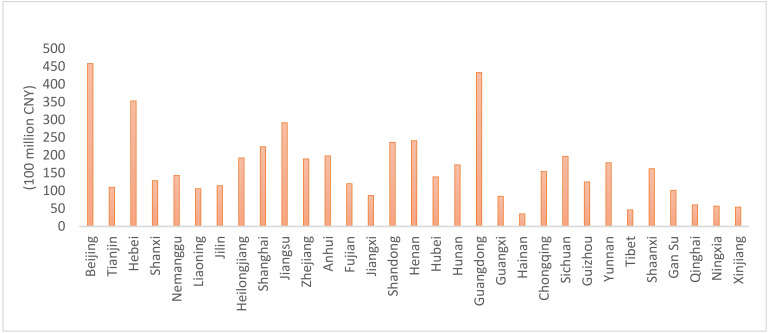
The local Chinese provincial fiscal expenditure on environmental protection (100 million CNY) in 2017.

**Table 1 ijerph-18-09667-t001:** The study variables.

Variable	Definition	Mesures	Ressources
COD	COD Emission in Waste Water	(10,000 tons)	National Bureau of Statistics of China, 2018
NOx	Ammonia Nitrogen Emission in Waste Water	(10,000 tons)	
SO_2_	Sulphur Dioxide Emission in Waste Gas	(10,000 tons)	
LGE	Local government expenditure	(100 million yuan)	
ENVE	provincial environmental expenditure	(100 million yuan)	
GDPP	Per Capita Gross Regional Product	(yuan/person)	
GDPP^2^	Per Capita Gross Regional Product square	(yuan/person)	
GDPP^3^	Per Capita Gross Regional Product cube	(yuan/person)	
SSVA	second-sector value added	(100 million yuan)	

**Table 2 ijerph-18-09667-t002:** Descriptive statistics of the study variables.

Variables	Mean	Median	Maximum	Minimum	Standard Deviation	Skewness	Kurtosis	Jarque–Bera	Sum	*p*-Value	Sum of Squares	Observation
(GDPP)	9.336	9.453	11.425	5.840	1.060	−0.802	3.634	42.357	3183.777	0.00	382.672	341
(COD)	3.738	3.887	5.289	0.405	0.942	−0.925	4.015	63.352	1274.882	0.00	302.316	341
(SO_2_)	7.362	8.006	10.416	−3.543	2.576	−2.011	7.520	520.294	2510.695	0.00	2257.490	341
(AN)	1.388	1.558	3.139	−1.966	0.973	−0.914	3.891	58.877	473.608	0.00	322.307	341
(LGE)	7.881	8.028	9.618	5.488	0.758	−0.640	3.310	24.683	2687.438	0.00	195.762	341
(ENVE)	4.302	4.411	6.127	1.562	0.786	−0.665	3.836	35.093	1467.277	0.002	210.492	341
(SSVA)	4.698	4.700	4.819	4.472	0.051	−0.378	3.498	11.679	1602.225	0.000	0.914	341

**Table 3 ijerph-18-09667-t003:** Results for the second-stage estimation: nexus between the three pollutant emissions factors SO_2_, COD, and AN emissions, and provincial economic growth per capita GDPP.

	(FMOLS)	Difference–GMM	Orthogonal–GMM
Ln(GDPP)_(−1)_		0.681 *	0.681 *
		32.917	11.563
Ln(LGE)	1.056 *	0.337 *	0.318 *
	11.104	14.197	6.007
Ln(ENVE)	−0.145 ***	−0.131 *	−0.103 *
	−1.666	−15.3904	−4.409
Ln(SO_2_)	0.025	−0.020	−0.018 *
	1.380	−6.536	−3.418
Ln(COD)	0.018	0.023 *	0.017 *
	0.245	6.511	2.337
Ln(AN)	0.360 *	−0.023 *	−0.016
	4.657	−2.891	−1.422
Ln(SSVA)	0.183 ***	0.844 *	0.696 *
	1.73	19.123	3.297
R-squared	0.880		
Adjusted R-squared	0.878		
S.E. of regression	0.362		
Long-run variance	0.220		
Root MSE		0.050	0.040
S.D. dependent var.		0.050	0.189
Sum squared resid.		0.708	0.463
Instrument rank		32	45

* and *** are significant at 0.01and 0.1, respectively. Note: SO_2_: Sulphur Dioxide Emission in Waste Gas; COD: COD Emission in Waste Water; AN: Ammonia Nitrogen Emission; LGE: Local government expenditure; ENVE: Provincial environmental expenditure; GDPP: Per Capita Gross Regional Product; SSVA: Second-sector value added. FMOLS: fully modified least square; GMM: generalized method of moments.

**Table 4 ijerph-18-09667-t004:** Results for the second-stage estimation using sulfur dioxide (SO_2_) as the dependent variable.

	(FMOLS)	Difference–GMM	Orthogonal–GMM
Ln(SO_2_)_(−1)_		1.290 ***	0.649 *
		21.8341	0.871
Ln(GDPP)	−11.919 **	−49.588	−2.375 *
	−2.235	−14.538	−0.752
Ln(GDPP)^2^	2.166 *	6.107	0.334 *
	3.5012	14.858	0.401
Ln(GDPP)^3^	−0.099 *	−0.249 **	−0.027 *
	−4.109	−15.594	−0.517
Ln(LGE)	−3.355 *	1.353 *	2.360 *
	−4.904	11.222	3.290
Ln(ENVE)	1.685 *	−0.408 ***	−0.372 *
	3.758	−9.731	−1.097
Ln(SSVA)	6.572 **	3.381 *	1.878 *
	2.144	21.087	0.290
R-squared	0.441		
Adjusted R-squared	0.382		
S.E. of regression	54.3194		
Long-run variance	31		
Root MSE		0.535	0.658
S.D. dependent var.		0.591	0.610
Sum squared resid.		79.9968	121.05
Instrument rank		32	31

*, **, and *** are significant at 0.01, 0.05, and 0.1, respectively. Note: SO_2_: Sulphur Dioxide Emission in Waste Gas; LGE: Local government expenditure; ENVE: Provincial environmental expenditure; GDPP: Per Capita Gross Regional Product; GDPP^2^: Per Capita Gross Regional Product square; GDPP^3^: Per Capita Gross Regional Product cube; SSVA: second-sector value added.

**Table 5 ijerph-18-09667-t005:** Results for the second-stage estimation using the chemical oxygen demand (COD) as the dependent variable.

	(FMOLS)	Difference–GMM	Orthogonal–GMM
Ln (COD)_(−1)_		0.092 *	0.040 *
		7.598	2.078
Ln (GDPP)	−4.256 *	−22.067 *	−0.626 *
	−2.615	−9.427	−0.822
Ln (GDPP)^2^	0.651 *	3.047 *	0.092 **
	3.447	10.141	3.725
Ln (GDPP)^3^	−0.026 *	−0.123 *	−0.003
	−3.608	−10.044	−4.310
Ln(LGE)	−0.452 **	−0.812 *	−0.532
	−2.164	−6.303	−0.595
Ln(ENVE)	0.249 ***	−0.210 *	0.258
	1.825	−11.741	−1.527
Ln(SSVA)	2.361 **	6.224 *	1.034 *
	2.524	12.453	6.593
R-squared	0.633		
Adjusted R-squared	0.627		
S.E. of regression	0.576		
Long-run variance	0.622		
Root MSE		0.492	0.400
S.D. dependent var		0.445	0.382
Sum squared resid		67.801	44.655
Instrument rank		32	32

*, **, and *** are significant at 0.01, 0.05, and 0.1, respectively. Note: COD: COD Emission in waste Water; LGE: Local government expenditure; ENVE: Provincial environmental expenditure; GDPP: Per Capita Gross Regional Product; GDPP^2^: Per Capita Gross Regional Product square; GDPP^3^: Per Capita Gross Regional Product cube; SSVA: Second-sector value added.

**Table 6 ijerph-18-09667-t006:** Results for the second-stage estimation using ammonia nitrogen emissions (AN) as the dependent variable.

	(FMOLS)	Difference–GMM	Orthogonal–GMM
Ln(AN)_it(−1)_		0.692 *	0.689 *
		19.364	21.051
Ln(GDPP)	−5.804 *	−49.093 *	−40.810 *
	−3.924	−4.432	−3.252
Ln(GDPP)^2^	0.823 *	5.588 *	4.642 *
	4.797	4.416	3.331
Ln(GDPP)^3^	−0.032 *	−0.215 *	−0.179 *
	−4.817	−4.564	−3.521
Ln(LGE)	−0.404 *	2.415 *	2.483 *
	−2.131	6.472	6.288
Ln(ENVE)	0.023 *	−0.916 *	−0.967 *
	0.187	−6.485	−6.920
Ln(SSVA)	2.845 *	8.882 *	9.337 *
	3.347	12.200	8.549
R-squared	0.664		
Adjusted R-squared	0.658		
S.E. of regression	0.569		
Long-run variance	0.514		
Root MSE		0.459	0.459
S.D. dependent var.		0.373	0.420
Sum squared resid.		59.027	58.871
Instrument rank		31	31

* is significant at 0.01. Note: AN: Ammonia Nitrogen Emission in Waste Water; LGE: Local government expenditure; ENVE: Provincial environmental expenditure; GDPP: Per Capita Gross Regional Product; GDPP^2^: Per Capita Gross Regional Product square; GDPP^3^: Per Capita Gross Regional Product cube; SSVA: Second-sector value added.

## Data Availability

The data presented in this study are openly available in https://data.stats.gov.cn.

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
