# Peer review of "Testing the Environmental Kuznets Curve Hypotheses in Chinese Provinces: A Nexus between Regional Government Expenditures and Environmental Quality"

_ijerph, 2021, doi:10.3390/ijerph18189667_

Round 1

Reviewer 1 Report

Dear Authors,

Although the manuscript is well and carefully organized, introductory matters are not properly treated. In this sense, it remains unclear to the reader whether the results are valid and useful or they might be worthless. Examples of this are:

  • The Introduction section is not well structured and a serious reorganization to make it up to the challenge is needed. For example just on the third line, one finds very confusing the statement: 'The Chinese government.........., has invested much in environmental and protection'. Unfortunately, being this the core matter in the study, one would expect a couple of paragraphs showing evidence of this, areas of development, etc, all that is needed in this regard. This is frustrating for the reader. Moreover, still, within the first paragraph another key area for the introductory stand referring to existing knowledge, i.e.' Certain researchers have recently concentrated their efforts on.....' it's not given the importance it has. In our view, the existing experience in factors relevant to the analysis needs to be addressed in much more detail in order to advance room for the current research while highlighting the importance of the results.
  • Following this reasoning, the Introductory section badly connects the vague preliminary elements with the N-shaped hypothesis. These links should be made clear to the reader only after a profound review of the introductory elements. Along the section, one realizes that vagueness remains and grows when general topics like the Paris Agreement are being brought up.
  • One main comment to the Methodology section: variables are not properly structured and a pre-screening analysis to understand the relevance of those variables into the study is missing. Why the three main variables have been chosen?. There seems not to be a clear understanding in the manuscript.

Author Response

Testing the Environmental Kuznets Curve Hypotheses in China Provinces: Nexus Between regional Government Expenditure and Environmental quality

Manuscript ID: ijerph-1308728

REVIEWER 1

Dear Authors,

Although the manuscript is well and carefully organized, introductory matters are not properly treated. In this sense, it remains unclear to the reader whether the results are valid and useful or they might be worthless. Examples of this are:

Response: Dear reviewer, we are very thankful for your constructive comments and positive response. We have entirely followed your suggestion and believe that the revised version is much improved.

  • The Introduction section is not well structured, and a serious reorganization to make it up to the challenge is needed. For example, just on the third line, one finds very confusing the statement: 'The Chinese government.........., has invested much in environmental and protection'. Unfortunately, being this the core matter in the study, one would expect a couple of paragraphs showing evidence of this, areas of development, etc, all that is needed in this regard. This is frustrating for the reader. Moreover, still, within the first paragraph another key area for the introductory stand referring to existing knowledge, i.e.' Certain researchers have recently concentrated their efforts on.....' it's not given the importance it has. In our view, the existing experience in factors relevant to the analysis needs to be addressed in much more detail to advance room for the current research while highlighting the importance of the results.

Response: Dear reviewer, thank you, as per your recommendations, the introduction has been formulated and thoroughly reorganized. Moreover, we have highlighted the importance of the study as well and answered why the three main variables were chosen.

  • Following this reasoning, the Introductory section badly connects the vague preliminary elements with the N-shaped hypothesis. These links should be made clear to the reader only after a profound review of the introductory elements. Along the section, one realizes that vagueness remains and grows when general topics like the Paris Agreement are being brought up.

Response: Dear Reviewer, thank you, we have now clearly mentioned the links among the underlying factors in this study. Now the introduction is much coherent. I hope it will meet your requirements.

  • One main comment to the Methodology section: variables are not properly structured and a pre-screening analysis to understand the relevance of those variables into the study is missing. Why the three main variables have been chosen?. There seems not to be a clear understanding in the manuscript.

Dear Reviewer, we are answering in the introduction the question Why the three main variables have been chosen?

Kindly have a look

In this study, three pollutants are used in the investigation to determine the effect that government spending has on environmental quality in China. This study uses three main determinants for the investigation, and these are (SO2), (COD), and (AN) emissions. The pollutants are used because they have different characteristics and can provide a detailed environmental quality analysis. For example, pollutants like soot and (SO2) emissions are recognized as air pollutants and have been under the government's control for a longer time due to their harm to the ecosystem and environment [14]. (SO2) emis-sions have contributed to the pollution problem in China in recent years[13]. This is also true for the (COD) emissions that indicated water pollution due to the combustion of coal [15]. China planned and managed to significantly reduce (COD) emissions by 8% within a period of six months through the 12th Five-Year Plan [13].

Dear reviewer, we are very grateful for your constructive comments, and I hope that making corrections to the manuscript, will make you satisfy you.

Best regards

Reviewer 2 Report

This paper tests the N-shaped EKC hypotheses in China provinces through the nexus between regional government expenditure and environmental quality. However, this paper does not meet the lowest criteria of publication.

  1. The authors should pay attention to the details of this paper. Some errors happened on the 1st page, such as “University of Castilla-La Mancha, Spian”, “Moreover, the study applied. The Generalized Method of Moments (GMM) was utilized to control the endogeneity potential”. There are some other errors on other pages. Please check this paper carefully.
  2. The accuracy of the research data is doubtful. According to common sense, economic growth per capita (GDPp) in Beijing should be much higher than that of Shandong, Henan, et al. However, the data characteristics of provincial economic growth per capital (GDPp) in Figure 1(d) is completely contrary to the facts.
  3. The Literature review is organized badly, which should be improved.
  4. The reason for choosing control variables is not given.
  5. The expression of models (1), (2), (3) and (4) should be improved by removing the ellipsis.
  6. The policy recommendation in the last part is omitted, which should be added.

Author Response

Testing environmental Kuznets Curve Hypotheses in China Provinces: Nexus Between regional Government Expenditure and Environmental quality

Manuscript ID: ijerph-1308728

REVIEWER 2

This paper tests the EKC hypotheses in China provinces through the nexus between regional government expenditure and environmental quality. However, this paper does not meet the lowest criteria of publication.

  1. The authors should pay attention to the details of this paper. Some errors happened on the 1st page, such as “University of Castilla-La Mancha, Spain”, “Moreover, the study applied. The Generalized Method of Moments (GMM) was utilized to control the endogeneity potential”. There are some other errors on other pages. Please check this paper carefully.

  • Response: Dear Reviewer, thank you, Based on your comments, previous errors on all manuscript pages have been corrected and

  • Kindly have a look

The study's estimate approach is a Generalized Method of Moments (GMM) model used to determine the direct and indirect impacts of provincial government expenditure. The study's econometric approach is the Generalized Method of Moments (GMM) technique. GMM is used to determine the direct and indirect impacts of provincial government Since the goal of the research is to distinguish the direct and indirect effects of the proportion of government expenditure over GDP, the estimation method utilized in this research is a two-stage regression model. Firstly, government expenditure on GDP per capita is examined, then on pollutant emissions, using the EKC framework. This uses both the GMM and Data fixed effects methods. The GMM approach was introduced by  [33] then developed by [33,34]. Previous studies have used the method to analyze the connection between GDP and pollutant emissions [35,36]. The results obtained from this research using the GGM method are used as the benchmark due to its efficiency in analyzing the data in different Chinese cities. Moreover, the first-difference GMM method can solve the endogeneity problem due to bilateral causality between GDP per capita and government spending[7,14].

The impacts of government expenditure on pollutant emissions are estimated using the (EKC) hypothesis empirical approach. Both the Fully Modified Ordinary Least Square (FMOLS) and the generalized method of moments (GMM) approach are used in the empirical estimations[38].  Introduced the Generalized Method of Moments (GMM), which was studied further by [32,33]. Because it can eliminate any endogeneity bias, the GMM method has been used in many previous studies to estimate the relationship between GDPP and pollutant emissions [32]. The results of the first-difference GMM technique are used as the benchmark in this study because individual differences may be eliminated [39]. It is reasonable to infer that there are significant differences between Chinese provinces, which the explanatory variables can highlight.

  1. The accuracy of the research data is doubtful. According to common sense, economic growth per capita (GDPp) in Beijing should be much higher than that of Shandong, Henan, et al. However, the data characteristics of provincial economic growth per capital (GDPp) in Figure 1(d) is completely contrary to the facts.
  • Response: Dear Reviewer, thank you, the validity of the data used in the study was confirmed and gave the same results even after they were presented in the form of charts.

  1. The Literature review is organized badly, which should be improved.
  • Response: Dear Reviewer, thank you. Based on your comments, the literature has been drafted and reviewed in a more thorough and organized structure.

  • Kindly have a look;

There have been several studies on the factors that influence environmental pollution up to now. Various studies, such as the Kuznets environmental curve and the hypothesis of a pollution refuge, are based on traditional theories explaining the relationship between environmental quality and economic growth [1,2]. Several previous studies have investigated the relationship between government expenditure and environmental quality in China. However, the results diverged significantly. The use, and then the excessive use and depletion of pollution, and climate change make up a broad field of empirical research in the literature pertaining to economic growth and the environment[20].moreover, explored the linkage between the government expenditure and pollution factors in china use GMM approach in the period 2002-2014, the study result indicates the existence of (EKC) hypothesis with inverted U-shaped and N-shaped [13].,a similar study by[21] analyzed the relationship between the of (SO2) emissions and government expenditures in the case of China in the period 2008-2013, applied spatial competition; the result confirmed the confirming a positive spatial correlation of (SO2) emissions among provinces in China. It also confirmed the existence of the N-shaped EKC hypothesis. The examination also indicates the positive association between the second sector value add and the (SO2) emissions.In the same stride, [22] clarified the relationship between environmental quality and economic growth in the context of the EKC hypothesis, used (SO2), (COD), emissions. The investigation result indicated a relationship between economic growth per capita and (SO2), (COD), emissions in the period 1998 to 2016 used also confirmed the pollution halo hypothesis. In the same context [2], examine the association relationship between noise pollution and the economic growth in 111 Chinese cities use the GMM approach to test the existence of the N-shaped EKC Hypothesis; the result confirmed the invert N-shaped between the economic growth per capita and the noise pollution.On the other hand, [23] investigating the linkage between the fiscal expenditure and (SO2) emissions as an environmental quality proxy in China provincial level, in the period 1995 to 2015, applied the GMM approach to check the role of provincial, regional economic growth on the financial autonomy in China. The results indicate an inerted-U shape between the economic growth per capita and the fiscal expenditure. The study also shows the positive relationship between economic growth and fiscal expenditure, also the inverted U-shaped and N-shaped relationship between the (SO2) emissions and economic growth in China. Moreover, the study also indicates that the second sector value adds have a positive effect on environmental degradation. Contrary to these results, [24] explored the association relationship between the local government expenditure and the (SO2) emission in 31china provinces from 2007 to 2016. The econometric approach reveals a negative relationship between (SO2) emission and local government expenditure. Also, the study in-dicates the existence of an inverted U-shaped (EKC) hypothesis in few prov-inces and not existent in others. [25], analyzed the relationship between the fiscal government and environmental quality in top ten Asian carbon emissions, in the period 1981 to 2018, the result shows the positive shock in government expenditure will worsen environmental quality in Malaysia, UAE, Thailand, Indonesia, Turkey, Iran, India, and China, and improve it in Japan. In the same context, the study by [26] estimated the GMM method to examine the relationship between local, provincial economic growth in 31 provinces in China and Pollutions factor's (SO2), (COD), and (AN) emissions in the period 2004 to 2016. the study result indicates the positive relationship between economic growth and the three pollutions factors. Inline the study by [27] when clarified the linkage relationship between the government expenditure as fiscal policy proxy and environmental quality in China in the period of 1980-2016, the empirical result indicates the positive relationship between the government expenditure and environmental degradation with the existence of (EKC) hypothesis both in the long and short run.[28] analyzed the linkage between the (SO2) emission and regional economic growth in the 29 provinces in the period 1998-2015. The empirical results indicated a positive relationship between the regional economic growth and the pollution factor the (SO2) emission, validating the inverted-U and the N-shaped EKC hypothesis. Moreover, in the context of Pakistan[29], when in-vestigating the nonlinear relationship between the role of fiscal policy on environmental quality, the empirical result shows the negative effect of fiscal policy instruments on the environmental quality in the long run, with the validation of the converted U-shaped relationship in the period 1985 to 2016. However, the analysis by [30] when analyzing the effect of both fiscal expenditure and environmental expenditure on the environmental quality in the emerging economies in the period 1995 to 2016, the study result indicate the positive effect of the environmental expenditure and environmental degradation and negative effect on the fiscal expenditure. Additionally, the empirical analysis by[31]when clarified the nexus between the public expenditure and environmental quality in 94 countries, which measures the environmental quality by the (SO2), and (AN) emissions in the period 1970 to 2008, the study demonstrates the positive and direct effect of significant of the public expenditure and environmental quality factor. The total effect was divided into direct and indirect effects, and both short-run and long-run impacts are discussed, respectively. In this research, the Environmental Kuznets Curve (EKC) hypothesis model is utilized to examine the impact of government expenditure on pollutant emissions.Thereafter, many researchers investigated the existence of an (EKC) hypothetical relationship between GDP per capita and certain pollutants [17, 18]. However, the estimation results are still controversial. [32] Found evidence for the existence of inverted-N shaped (EKC) hypothesis for (COD), emissions for the whole of China and Shanghai city, respectively. [32]also reported the existence of inverted-U shaped EKC in the G20 group members.Additionally, [13] dewed on Chinese survey data and found that the relationship between (CO2) emissions per capita and GDP per capita is N shaped, suggesting there might not be a turning point of (CO2) emissions in China. As many previous studies have utilized typical pollutants as the proxy of environmental quality [13] in this study, three representative pollutants(SO2), (COD), and (AN) emissions are chosen for the empirical estimations. These different pollutants have various characteristics and therefore reflect the different aspects of China's environmental pollution. Consequently, examining these pollutants helps us understand the full story of the environmental impacts of government spending in China.Despite several previous studies, the link between government expenditure and environmental quality in China is rarely explored, mainly when using provincial data. As a result, this research analyses panel data from 31 Chinese provinces to examine the link between government expenditure and certain pollutants (SO2), (COD), and (AN) emissions, as well as to estimate the total effect, as well as the effects of government expenditure.

  1. The reason for choosing control variables is not given.

More clarify was added on the motive for choosing the study variables, both in the introduction and data description paragraph

  • Kindly have a look

In this study, three pollutants are used in the investigation to determine the effect that government spending has on environmental quality in China. This study uses three main determinants for the investigation, and these are (SO2), (COD), and (AN) emissions. The pollutants are used because they have different characteristics and can provide a detailed environmental quality analysis. For example, pollutants like soot and (SO2) emissions are recognized as air pollutants and have been under the government's control for a longer time due to their harm to the ecosystem and environment [14]. (SO2) emissions have contributed to the pollution problem in China in recent years[13]. This is also true for the (COD) emissions that indicated water pollution due to the combustion of coal [15]. China planned and managed to significantly reduce (COD) emissions by 8% within a period of six months through the 12th Five-Year Plan [13].

  1. The expression of models (1), (2), (3) and (4) should be improved by removing the ellipsis.

  • Response: Dear Reviewer, thank you, we correct and progress the grammatical structure of the result and the model by removing the ellipsis.
  1. The policy recommendation in the last part is omitted, which should be added.
  • Response: Dear Reviewer, thank you, the conclusion was further developed by adding recommendations.
  • Kindly have a look

Based on the conclusions and preliminary results, we can give the following recommendation first: Increasing attention to the local environmental system of the Chinese provinces and working on designing a system that balances local economic growth and preserving the environment. It is also necessary to work on regulating local regulations and laws to cover various local environmental dimensions such as gaseous and industrial emissions and water pollution resulting from sewage. Sustainable. Second: Working on building a local government spending system on foundations based on the foundations of governance and monitoring the independence of government spending for local districts to maximize the volume of environmental conservation and reduce emissions. Third: Improving the industrial structure and accelerating the transformation of the local industrial structure into an environment-friendly system. Although the secondary industry was a significant growth engine for China's economy, it has caused growing environmental pollution, which requires the intervention of the Chinese government to develop the industrial sector and achieve sustainable environmental development goals. Fourth: Decision-makers should build strategies to preserve the environmental system on the most basic assumptions that keep pace with the size of the Chinese economy and the size of the environmental challenges it faces, such as the Keynesian hypotheses, which depend on the cube of economic growth, which predicts the existence of the hypothesis of the existence of a third stage that allows a return of the parallel between economic growth and environmental pollution when giving as the inverted N-shaped, in contrast to the classical EKC Hypothesis, which stops at squaring the volume of economic growth, which explains the inverse relationship between economic growth and the importance of pollution and give as the inverted U-shaped. This study encountered some limitations and guidelines for future research.1) This study is limited to 31 Chinese provinces with further economic growth. 2) Furthermore, this study uses data from 2007 to 2017, and the period for future studies can be extended by studies conducted in the future. 3) Other than that, future studies could also include an interactive term for energetic innovation and public investments in the environment. 4) Finally, the role of renewable energies and the governance criterion can also be covered in studies to be conducted in future studies.

Dear reviewer, we are very grateful for your constructive comments, and I hope that making corrections to the manuscript, will make you satisfy you.

Best regards

Reviewer 3 Report

The aim of this study was to evaluate the effects of various factors, including the government expenditures, economic growth per capita, environment protection expenditure, and second sector value added to the Environment. This study is one of the few that address the effects of the emissions of the SO2, COD, and NOX on economic growth. However, I am afraid some terms were misused in this study, such as the chemical oxygen demand and the ammonia nitrogen. Furthermore, the manuscript also requires an extensive editing of the English language. See my comments below:

The manuscript does not have line numbers, which makes it difficult to provide detailed reviews.

Abstract: How can the COD be emitted? The chemical formula of ammonia Nitrogen is not NOx, NOX refers to a family of poisonous highly reactive gases. While the COD indicates the measure of the amount of oxygen that can be consumed by reactions in a measured solution.

Furthermore, the term emission should be used for the discharge of gaseous substances, not the COD, which is contained in solutions. Perhaps an alternative to the COD, in terms of emission, would have been the emission of carbon dioxide (CO2), which could be relatable to the level of industrialization of a country.

Page 2: What does EKC stand for? Please provide a definition before using the acronym throughout the text.

Page 3: “where we note in return that areas with chemical pollution record significant economic growth, except for Beijing and Beijing” Beijing is repeated twice here, is this a typo?

Page 3: “We also note that dioxide emissions have increased in the northwestern and north-eastern regions, which are areas rich in mining and metallurgical industry [22].” What do the authors referred to by dioxide here?

Page 3: Check the grammar, please rephrase this sentence: “However, on the other hand, we note a low rate of economic growth for the region, and this is due to several reasons as they are considered areas that are transported Raw materials from them are transported to other areas that contribute to local heavy industries or exports, which makes the ban not benefit from them.”

Figure 2: Please add error bars to the histogram. What is the variable displayed in the y-axis? What are the relevant units?

Page 4: This is a long run sentence, please amend it accordingly: “Based to figure two, when the chart shows china provinces with government expenditure on the protection of the local environment, moreover, the figure showed two levels, the first level is the provinces that have great interest and price in preserving the environment by raising the volume of spending on the environment, such as Beijing, Jiangsu and Guandong, similar to the Hubei region, which is known for heavy industries with high pollution and its proximity to Beijing, which led the local government to work more to improve the quality of the environment, On the other hand, some areas have a weak interest in the environment, and where, as shown, the areas with high economic growth pay more attention to the environment than countries with weak economic growth.”

 Furthermore, Figure 2 doesn’t highlight the two sections referred to in this paragraph.

Page 4: At the end of Page 4 and the beginning of page 5, the authors comments the outcome of some studies. One easy way to highlight such studies is to tabulate the findings and the parameters of the relevant studies. This enables an easy comparison and fluid discussion thereof.

Page 5: The section model development is not clearly substantiated.

Page 6: What was Equation 1 derived from. Furthermore, the three dots at the end of the equation suggests that it includes other parameters, is that case? Can a generalized form of Equations 1, 2, 3, and 4 be used?

Page 7: This dln(GDP) dx ∗ dln(SO2) dln(GDP) should be included in the text as a separate equation.

Page 7: The emission of COD is once again referred to here: “ Ammonia nitrogen emissions (10,000 tons), sulphur dioxide (SO2) (10,000 tons) and chemical oxygen demand (COD) total emissions of the four main air pollutants nitrogen oxides (NOx).”

Table 1: It could be interesting to visualize probability density functions of the parameters tabulated in this table to provide a more insightful representation of the information provided.

Page 7: below should not be used to refer to a figure or a table in the text, see here: “. However, as seen in Tables bellow, the results are inconclusive between classifications and the different methods used. The (FMOLS)- Difference-GMM Orthoonal-GMM.There are two stages to the estimating technique [38,39]. “

Author Response

Testing the Kuznets Curve Hypotheses in China Provinces: Nexus Between regional Government Expenditure and Environmental quality

Manuscript ID: ijerph-1308728

REVIEWER 3

The aim of this study was to evaluate the effects of various factors, including the government expenditures, economic growth per capita, environment protection expenditure, and second sector value added to the Environment. This study is one of the few that address the effects of the emissions of SO2, COD, and NOon economic growth. However, I am afraid some terms were misused in this study, such as the chemical oxygen demand and the ammonia nitrogen. Furthermore, the manuscript also requires extensive editing of the English language. See my comments below:

The manuscript does not have line numbers, which makes it difficult to provide detailed reviews.

Response: Dear Reviewer, thank you, the lines were added to the manuscript.

Abstract: How can COD be emitted? The chemical formula of ammonia Nitrogen is not NOx, NOX refers to a family of poisonous highly reactive gases. While the COD indicates the measure of the amount of oxygen that can be consumed by reactions in a measured solution.

Furthermore, the term emission should be used for the discharge of gaseous substances, not the COD, which is contained in solutions. Perhaps an alternative to the COD, in terms of emission, would have been the emission of carbon dioxide (CO2), which could be relatable to the level of industrialization of a country.

Page 2: What does EKC stand for? Please define using the acronym throughout the text.

  • Response: Dear Reviewer, thank you. We have first defined the full form of abbreviation in the text then we used its abbreviation for the whole paper as per journal policy.

Kindly have a look

The Environmental Kuznets Curve (EKC) hypothesis explains an inverted N-shaped relationship between economic growth and environmental degradation environmental pressure increases in the early stages of economic growth due to the increased release of pollutants and the extensive and intensive exploitation of natural resources associated with the greater use of production resources and the adoption of certain production technologies for the growing economic activities, up to a certain level, as income rises; and after that, it decreases, probably because of the growing public awareness and concern about environmental degradation and the research and development activities being oriented more toward the concept of the green economy when GDP grows at a high-level the third stage, which can return to the parallel relationship between economic growth and pollutants, so we use the cube of economic growth. [1,2, 20]

Page 3: “where we note in return that areas with chemical pollution record significant economic growth, except for Beijing and Beijing” Beijing is repeated twice here, is this a typo?

  • Response: Dear Reviewer, thank you Yes, is typo errors, referring to Beijing. This has been corrected in the same paragraph.

Page 3: “We also note that dioxide emissions have increased in the northwestern and north-eastern regions, which are areas rich in mining and metallurgical industry [22].” What do the authors refer to by dioxide here?

  • Response: Dear Reviewer, thank you. We mean oxide specifically sulfur dioxide (SO2), it was corrected. 

Page 3: Check the grammar, please rephrase this sentence: “However, on the other hand, we note a low rate of economic growth for the region, and this is due to several reasons as they are considered areas that are transported Raw materials from them are transported to other areas that contribute to local heavy industries or exports, which makes the ban not benefit from them.”

  • Response: Dear Reviewer, thank you. The grammar check has been carried out and the above paragraph has been redrafted

Figure 2: Please add error bars to the histogram. What is the variable displayed in the y-axis? What are the relevant units?

  • Response: Dear Reviewer, thank you. The relevant unit has been added (100 million yuan)

Page 4: This is a long run sentence, please amend it accordingly: “Based to figure two, when the chart shows china provinces with government expenditure on the protection of the local environment, moreover, the figure showed two levels, the first level is the provinces that have great interest and price in preserving the environment by raising the volume of spending on the environment, such as Beijing, Jiangsu and Guandong, similar to the Hubei region, which is known for heavy industries with high pollution and its proximity to Beijing, which led the local government to work more to improve the quality of the environment, On the other hand, some areas have a weak interest in the environment, and where, as shown, the areas with high economic growth pay more attention to the environment than countries with weak economic growth.” Furthermore, Figure 2 doesn’t highlight the two sections referred to in this paragraph.

  • Response: Dear Reviewer, thank you

This section meaning was corrected and the mean is that there are two levels of expenditure policy, counties with high expenditure on the environment, which makes the quality of their environment good, and there are areas with low expenditure on the environment with a lower quality environment than the first have higher environmental expenditures.

Page 4: At the end of Page 4 and the beginning of page 5, the authors comment on the outcome of some studies. One easy way to highlight such studies is to tabulate the findings and the parameters of the relevant studies. This enables an easy comparison and fluid discussion thereof.

  • Response: Dear Reviewer, thank you. The section has been rephrased and corrected
  • Kindly have a look;

There have been several studies on the factors that influence environmental pollution up to now. Various studies, such as the Kuznets environmental curve and the hypothesis of a pollution refuge, are based on traditional theories explaining the relationship between environmental quality and economic growth [1,2]. Several previous studies have investigated the relationship between government expenditure and environmental quality in China. However, the results diverged significantly. The use, and then the excessive use and depletion of pollution, and climate change make up a broad field of empirical research in the literature pertaining to economic growth and the environment[20].moreover, explored the linkage between the government expenditure and pollution factors in china use GMM approach in the period 2002-2014, the study result indicates the existence of (EKC) hypothesis with inverted U-shaped and N-shaped [13].,a similar study by[21] analyzed the relationship between the (SO2) emissions and government expenditures in the case of China in the period 2008-2013, applied spatial competition; the result confirmed the confirming a positive spatial correlation of (SO2) emissions among provinces in China. It also confirmed the existence of the N-shaped EKC hypothesis. The examination also indicates the positive association between the second sector value add and the (SO2) emissions.In the same stride, [22] clarified the relationship between environmental quality and economic growth in the context of the EKC hypothesis, used (SO2), (COD), emissions. The investigation result indicated a relationship between economic growth per capita and (SO2), (COD), emissions in the period 1998 to 2016 used also confirmed the pollution halo hypothesis. In the same context [2], examine the association relationship between noise pollution and the eco-nomic growth in 111 Chinese cities use the GMM approach to test the existence of the N-shaped EKC Hypothesis; the result confirmed the invert N-shaped between the economic growth per capita and the noise pollution.On the other hand, [23] investigating the linkage between the fiscal expenditure and (SO2) emissions as an environmental quality proxy in China provincial level, in the period 1995 to 2015, applied the GMM approach to check the role of provincial, regional economic growth on the financial autonomy in China. The results indicate an inerted-U shape between the economic growth per capita and the fiscal expenditure. The study also shows the posi-tive relationship between economic growth and fiscal expenditure, also the inverted U-shaped and N-shaped relationship between the (SO2) emissions and economic growth in China. Moreover, the study also indicates that the second sector value adds have a positive effect on environmental degradation. Contrary to these results, [24] explored the association relationship between the local government expenditure and the (SO2) emission in 31china provinces from 2007 to 2016. The econometric approach reveals a negative relationship between (SO2) emission and local government expenditure. Also, the study indicates the existence of an inverted U-shaped (EKC) hypothesis in few provinces and not existent in others. [25], analyzed the relationship between the fiscal government and environmental quality in top ten Asian carbon emissions, in the period 1981 to 2018, the result shows the positive shock in government expenditure will worsen environmental quality in Malaysia, UAE, Thailand, Indonesia, Turkey, Iran, India, and China, and improve it in Japan. In the same context, the study by [26] estimated the GMM method to examine the relationship between local, provincial economic growth in 31 provinces in China and Pollutions factor's (SO2), (COD), and (AN) emissions in the period 2004 to 2016. the study result indicates the positive relationship between economic growth and the three pollutions factors. Inline the study by [27] when clarified the linkage relationship between the government expenditure as fiscal policy proxy and environmental quality in China in the period of 1980-2016, the empirical result indicates the positive relationship between the government expenditure and environmental degradation with the existence of (EKC) hypothesis both in the long and short run.[28] analyzed the linkage between the (SO2) emission and regional economic growth in the 29 provinces in the period 1998-2015. The empirical results indicated a positive relationship between the regional economic growth and the pollution factor the (SO2) emission, validating the inverted-U and the N-shaped EKC hypothesis. Moreover, in the context of Pakistan[29], when in-vestigating the nonlinear relationship between the role of fiscal policy on envi-ronmental quality, the empirical result shows the negative effect of fiscal policy instruments on the environmental quality in the long run, with the validation of the converted U-shaped relationship in the period 1985 to 2016. However, the analysis by [30] when analyzing the effect of both fiscal expenditure and environmental expenditure on the environmental quality in the emerging economies in the period 1995 to 2016, the study result indicate the positive effect of the environmental expenditure and environmental degradation and negative effect on the fiscal expenditure. Additionally, the empirical analysis by[31]when clarified the nexus between the public expenditure and environmental quality in 94 countries, which measures the environmental quality by the (SO2), and (AN) emissions in the period 1970 to 2008, the study demonstrates the positive and direct effect of significant of the public expenditure and environmental quality factor. The total effect was divided into direct and indirect effects, and both short-run and long-run impacts are discussed, respectively. In this research, the Environmental Kuznets Curve (EKC) hypothesis model is utilized to examine the impact of government expenditure on pollutant emissions.Thereafter, many researchers investigated the existence of an (EKC) hypothetical relationship between GDP per capita and certain pollutants [17, 18]. However, the estimation results are still controversial. [32] Found evidence for the existence of inverted-N shaped (EKC) hypothesis for (COD), emissions for the whole of China and Shanghai city, respectively. [32]also reported the existence of inverted-U shaped EKC in the G20 group members.Additionally, [13] dewed on Chinese survey data and found that the relationship between (CO2) emissions per capita and GDP per capita is N shaped, suggesting there might not be a turning point of (CO2) emissions in China. As many previous studies have utilized typical pollutants as the proxy of environmental quality [13] in this study, three representative pollutants(SO2), (COD), and (AN) emissions are chosen for the empirical estimations. These different pollutants have various characteristics and therefore reflect the different aspects of China's environmental pollution. Consequently, examining these pollutants helps us understand the full story of the environmental impacts of government spending in China.Despite several previous studies, the link between government expenditure and environmental quality in China is rarely explored, mainly when using provincial data. As a result, this research analyses panel data from 31 Chinese provinces to examine the link between government expenditure and certain pollutants (SO2), (COD), and (AN) emissions, as well as to estimate the total effect, as well as the effects of government expenditure.

Page 5: The section model development is not substantiated.

  • Response: Dear Reviewer, thank you. The study model has been reformulated, making it simple and clear as shown in the manuscript.

Page 6: What was Equation 1 derived from? Furthermore, the three dots at the end of the equation suggests that it includes other parameters, is that case? Can a generalized form of Equations 1, 2, 3, and 4 be used?

  • Response: Dear Reviewer, thank you for the three 1, 2, 3, and 4 equations that have been combined into one equation(1)

Page 7: This dln(GDP) dx ∗ dln(SO2) dln(GDP) should be included in the text as a separate equation.

  • Response: Dear Reviewer, thank you The model has been reformulated, and the above equation has been deleted, which no longer clearly expresses the model

Page 7: The emission of COD has once again referred to here: “ Ammonia nitrogen emissions (10,000 tons), sulphur dioxide (SO2) (10,000 tons) and chemical oxygen demand (COD) total emissions of the four main air pollutants nitrogen oxides (NOx).”

  • Response: Dear Reviewer, thank you. This has been deleted from the paragraph and added in the data descriptive

Table 1: It could be interesting to visualize probability density functions of the parameters tabulated in this table to provide a more insightful representation of the information provided.

  • Response: Dear Reviewer, thank you. The probability values have been added to the first table

Kindly have a look

                                               Table 1. Study variables statistics descriptive.

Variabeles

Mean

Median

Maximum

Minimum

Std. Dev.

Skewness

Kurtosis

Jarque-Bera

Sum

Probability

Sum Sq. Dev.

Obs

(GDPP)

9.336

9.453

11.425

5.840

1.060

-0.802

3.634

42.357

3183.777

0.00

382.672

341

(COD)

3.738

3.887

5.289

0.405

0.942

-0.925

4.015

63.352

1274.882

0.00

302.316

341

(SO2)

7.362

8.006

10.416

-3.543

2.576

-2.011

7.520

520.294

2510.695

0.00

2257.490

341

(AN)

1.388

1.558

3.139

-1.966

0.973

-0.914

3.891

58.877

473.608

0.00

322.307

341

(LGE)

7.881

8.028

9.618

5.488

0.758

-0.640

3.310

24.683

2687.438

0.00

195.762

341

(ENVE)

4.302

4.411

6.127

1.562

0.786

-0.665

3.836

35.093

1467.277

0.002

210.492

341

(SSVA)

4.698

4.700

4.819

4.472

0.051

-0.378

3.498

11.679

1602.225

0.00

0.914

341

Page 7: below should not be used to refer to a figure or a table in the text, see here: “. However, as seen in the Tables below, the results are inconclusive between classifications and the different methods used. The (FMOLS)- Difference-GMM Orthoonal-GMM.There are two stages to the estimating technique [38,39]. “

  • Response: Dear Reviewer, thank you The paragraph has been deleted and replaced with an appropriate and more consistent paragraph

Kindly have a look

The study's estimate approach is a Generalized Method of Moments (GMM) model used to determine the direct and indirect impacts of provincial government expenditure. The study's econometric approach is the Generalized Method of Moments (GMM) technique. GMM is used to determine the direct and indirect impacts of provincial government since the goal of the research is to distinguish the direct and indirect effects of the proportion of government expenditure over GDP, the estimation method utilized in this research is a two-stage regression model. Firstly, government expenditure on GDP per capita is examined, then on pollutant emissions, using the EKC framework. This uses both the GMM and Data fixed effects methods. The GMM approach was introduced by [33] then developed by [33,34]. Previous studies have used the method to analyze the connection between GDP and pollutant emissions [35,36]. The results obtained from this research using the GGM method are used as the benchmark due to its efficiency in analyzing the data in different Chinese cities. Moreover, the first-difference GMM method can solve the endogeneity problem due to bilateral causality between GDP per capita and government spending[7,14]. The impacts of government expenditure on pollutant emissions are estimated using the (EKC) hypothesis empirical approach. Both the Fully Modified Ordinary Least Square (FMOLS) and the generalized method of moments (GMM) approach are used in the empirical estimations [38]. Introduced the Generalized Method of Moments (GMM), which was studied further by [32,33]. Because it can eliminate any endogeneity bias, the GMM method has been used in many previous studies to estimate the relationship between GDPP and pollutant emissions [32]. The results of the first-difference GMM technique are used as the benchmark in this study because individual differences may be eliminated [39]. It is reasonable to infer that there are significant differences between Chinese provinces, which the explanatory variables can highlight.

Dear reviewer, we are very grateful for your constructive comments, and I hope that making corrections to the manuscript, will make you satisfy you.

Best regards

Round 2

Reviewer 1 Report

Unfortunately, the authors have acted rather hastily to deliver an unsatisfactory version of the manuscript.

Quick examples of this (not the only ones) are:

a) Comment made before not resolved: For example, just on the third line, one finds very confusing the statement: 'The Chinese government.........., has invested much in environmental and protection'. Unfortunately, being this the core matter in the study, one would expect a couple of paragraphs showing evidence of this, areas of development, etc, all that is needed in this regard. This is frustrating for the reader. 

No clarification or references supporting the statement have been included.

b) Reference [5] before missing in the text although included in the list of references as [37], i.e. S. S. Wang, D. Q. Zhou, P. Zhou, and Q. W. Wang, “CO2 emissions, energy consumption and economic growth in Chipanel data analysis,” Energy Policy, vol. 39, no. 9, pp. 4870–4875, 2011, doi: 10.1016/j.enpol.2011.06.032. 

c) Comment made before not resolved. Why the three main variables have been chosen?. There seems not to be a clear understanding in the manuscript

We can't see where in the manuscript, clarification or references supporting the statement have been included.

Author Response

" Please see the attachment."

Reviewer 2 Report

It's suitable to publish.

Reviewer 3 Report

The quality of the manuscript has significantly improved. However, I would suggest considering the following comments to make it even better:

In the abstract, the authors defined the COD as the chemical oxygen demand, which, in environmental chemistry, is the measure of the amount of oxygen required to oxidize chemical compounds contained in solutions such as wastewater, not gaseous pollutants. However, the authors in multiple instances refer to the COD as a substance that is emitted, therefore understood as being released in the air (see lines 90, 96, 97). The use of emission in these sentences can be confusing and misleading, I would recommend using words such as discharge or release to reduce the ambiguity of these sentences as it relates to COD.  

Figure 2: are the values reported here averaged or tied to a specific year. If tied to a specific year or season, please specify; and if the values are averaged, please add error bars to the histogram.

Why are the acronyms put in parentheses throughout the text after being defined? I believe it should be fine to remove the parentheses around these acronyms after they have been defined.

Equations 1 and 2: Please provide the complete list of the meaning of the parameters in each equation.

Table 1: does the probability in this table refer to the p-value? If that is the case, then it should be more appropriate to use p-value instead of the term probability

Round 3

Reviewer 1 Report

Dear authors, Appreciated the efforts but changes/additions were made in great haste. The manuscript needs a complete overhaul.

Author Response

Dear reviewer, we have improved the article as per your recommendations. We made major improvements that makes the article more interesting and good. We hope it will meet your requirements .

Best regards
